# Emergent ferromagnetism with superconductivity in Fe(Te,Se) van der Waals Josephson junctions

Gang Qiu [1,11] ✉, Hung-Yu Yang [1,11], Lunhui Hu[2], Huairuo Zhang [3,4], Chih-Yen Chen [5], Yanfeng Lyu[6], Christopher Eckberg [1,7,8,9], Peng Deng [1,10], Sergiy Krylyuk[3], Albert V. Davydov [3], Ruixing Zhang [2] & Kang L. Wang [1] ✉

Ferromagnetism and superconductivity are two key ingredients for topological superconductors, which can serve as building blocks of fault-tolerant quantum computers. Adversely, ferromagnetism and superconductivity are typically also two hostile orderings competing to align spins in different configurations, and thus making the material design and experimental implementation extremely challenging. A single material platform with concurrent ferromagnetism and superconductivity is actively pursued. In this paper, we fabricate van der Waals Josephson junctions made with iron-based superconductor Fe(Te,Se), and report the global device-level transport signatures of interfacial ferromagnetism emerging with superconducting states for the first time. Magnetic hysteresis in the junction resistance is observed only below the superconducting critical temperature, suggesting an inherent correlation between ferromagnetic and superconducting order parameters. The 0-π phase mixing in the Fraunhofer patterns pinpoints the ferromagnetism on the junction interface. More importantly, a stochastic field-free superconducting diode effect was observed in Josephson junction devices, with a significant diode efficiency up to 10%, which unambiguously confirms the spontaneous time-reversal symmetry breaking. Our work demonstrates a new way to search for topological superconductivity in iron-based superconductors for future high $T_c$ fault-tolerant qubit implementations from a device perspective.

By virtue of its immunity to adiabatic perturbation, the topological quantum computer is deemed a promising paradigm for fault-tolerant quantum information processing. The mainstream hardware approach to implement topological qubits is to utilize non-Abelian quasiparticle excitations – Majorana fermions in topological superconductors (TSCs)[1–3]. Unfortunately, the two key ingredients to create TSCs, ferromagnetism and superconductivity are generally two hostile orderings that compete to arrange electron spins in different configurations. The spins in a ferromagnetic material are aligned in parallel due to exchange interactions. In contrast Cooper pairs in a

[1]Department of Electrical and Computer Engineering, University of California, Los Angeles, CA 90095, USA. [2]Department of Physics & Astronomy, The University of Tennessee, Knoxville, Knoxville, TN 37996, USA. [3]Materials Science and Engineering Division, National Institute of Standards and Technology (NIST), Gaithersburg, MD 20899, USA. [4]Theiss Research, Inc, La Jolla, CA 92037, USA. [5]Department of Electrophysics, National Yang Ming Chiao Tung University (NYCU), Hsinchu 30010, Taiwan. [6]School of Science, Nanjing University of Posts and Telecommunications, 210023 Nanjing, China. [7]Fibertek Inc, Herndon, VA 20171, USA. [8]DEVCOM Army Research Laboratory, Adelphi, MD 20783, USA. [9]DEVCOM Army Research Laboratory, Playa Vista, Los Angeles, CA 90094, USA. [10]Beijing Academy of Quantum Information Sciences, 100193 Beijing, China. [11]These authors contributed equally: Gang Qiu, Hung-Yu Yang. ✉e-mail: gqiu@g.ucla.edu; wang@ee.ucla.edu

mundane s-wave superconductor feature singlet-pairing states of two electrons with opposite spins. In carefully engineered heterostructures, the magnetism and superconductivity orderings can be mixed through the proximity effect[4]. Yet, a single material with inherent coexistence of ferromagnetism and superconductivity is rare to encounter in nature, which would favorably ease the fabrication challenge compared to other proposed hybrid topological superconductor implementations[5–9].

Recently, the iron-based superconductor Fe(Te,Se) (FTS) stands out as a promising intrinsic TSC candidate with the coexistence of topological Dirac surface states, bulk superconductivity, and ferromagnetism. The bulk s-wave superconductivity induces a topological superconducting gap at the Dirac surfaces states below the superconducting $T_c$[10], which can be directly visualized by high-resolution angle-resolved photon-emission spectroscopy (ARPES) experiments[11]. Traces of Majorana bound states (MBS) in vortex cores were also detected using tunneling spectroscopy measurements[12,13]. In these earlier works, an external magnetic field was still required to observe MBS. Recently, evidence of the spontaneous time-reversal symmetry breaking on the surface of this material has been reported via several techniques, including ARPES[14], nitrogen-vacancy (NV) center imaging[15], surface magneto-optic Kerr effect (MOKE)[16], and nano superconducting quantum interference device (SQUID) imaging[17]. This superconductivity-compatible surface magnetism is unexpected, and the mechanism is not yet clear. And the current studies of ferromagnetism in superconducting FTS using spectroscopy and imaging

techniques are subject to material inhomogeneity, magnetic impurities, and other trivial origins, such as Caroli-de Gennes Matricon states[18]. Device level transport measurement is yet to be explored, which can eliminate spatial variations by measuring averaged global topological features. In addition, understanding mesoscopic transport behavior is critical for top-down scalable manufacturing of topological qubits. In this work, we fabricate and characterize FTS van der Waals Josephson junction (vJJ) devices and observed the unconventional interfacial ferromagnetism emerging with superconductivity unambiguously with the following major observations: (1) a magnetic hysteresis loop (R vs. H) below superconducting Tc; (2) a π phase component in the Fraunhofer patterns of the vJJ arising from ferromagnetism; (3) the stochastic supercurrent diode effect without external magnetic field which suggests spontaneous time-reversal symmetry breaking. This is the first time that evidence of time-reversal symmetry-breaking superconductivity in FTS has been reported through device transport experiments. Our work provides a unique way of identifying time-reversal symmetry breaking in superconductivity in topological superconductors candidates.

## FTS vdW Josephson junctions
We fabricate Josephson junctions by employing the vdW gap between two homogeneously stacked FTS flakes, as shown in Fig. 1a, b. Two FTS flakes are exfoliated from a single crystal Fe(Te$_{0.58}$Se$_{0.42}$) onto a SiO$_2$/Si substrate, then capped with a hexagonal-boron nitride (h-BN) flake and transferred onto pre-patterned electrodes. More details of material

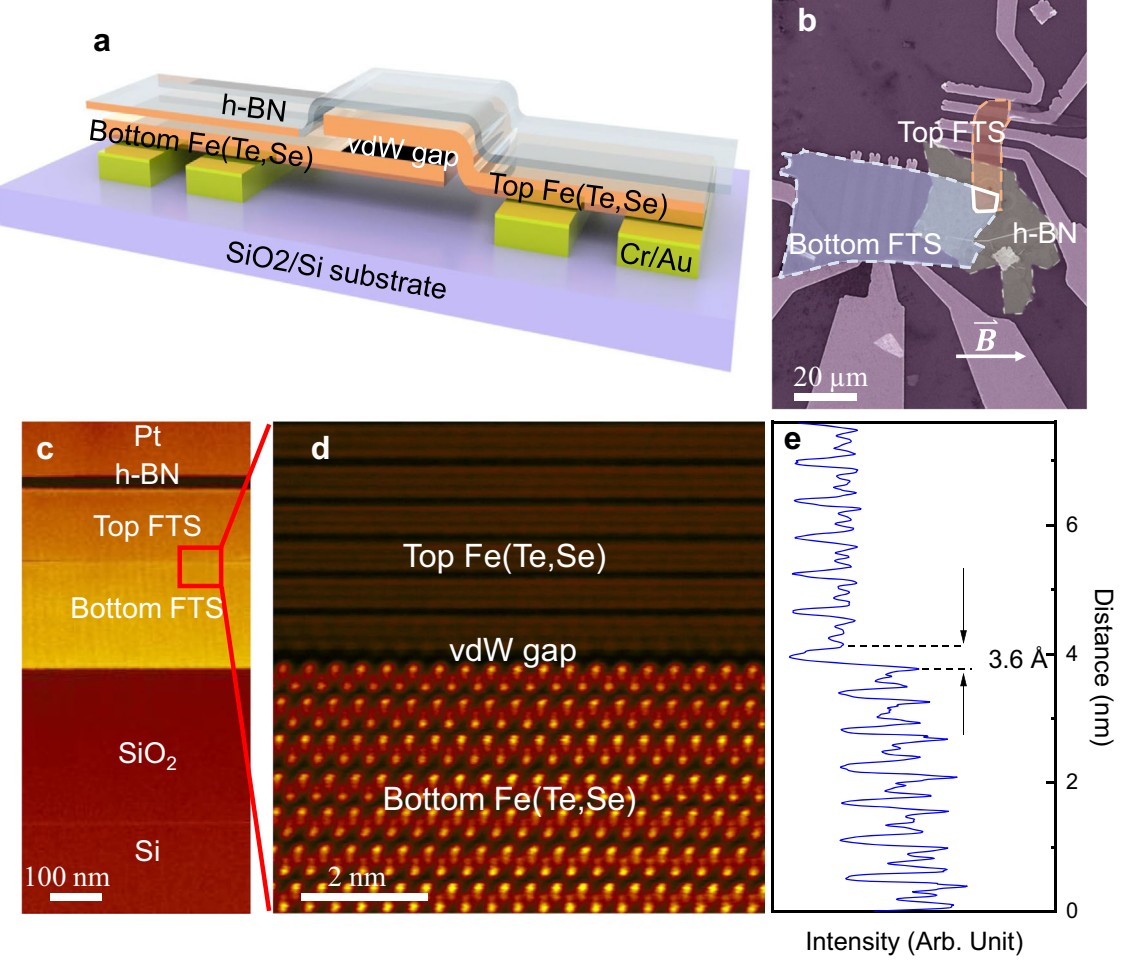

**Fig. 1 | Fe(Te,Se) van der Waals Josephson junctions (FTS vJJ). a** The device schematic of an FTS vJJ. **b** A false-colored scanning electron microscopy image of an FTS vJJ device. **c** Low-magnification ADF-STEM image showing the cross-section of a vJJ stack. **d** atomic resolution image from the region defined by a red box in (**c**) showing a van der Waals gap between the two FTS flakes. **e** Integrated line intensity from STEM image in (**d**). An interatomic van der Waals gap of 3.6 Å is determined.

synthesis and device fabrication are available in the Methods section and Supplementary Note 1. Atomic resolution annular dark-field scanning transmission electron microscopy (ADF-STEM) imaging of the cross-sectional sample (Fig. 1c, d) reveals an interatomic vdW gap of 3.6 Å between the two FTS flakes, as shown in Fig. 1e (see "Methods" section for more details). This vdW gap is sufficient to establish a phase separation between two adjacent superconducting flakes to form a superconductor-vdW-superconductor Josephson junction[19,20].

The Josephson effect is confirmed by detecting a supercurrent flowing across the junction as demonstrated in the R-T curve in Fig. 2a. The FTS superconducting flakes (blue curves) show a critical temperature $T_c$ of 14.5 K, in accordance with the value reported in the literature[21]. The resistance of superconducting flakes (reservoirs) completely vanishes at 14 K (blue curve in Fig. 2a); therefore, the remnant resistance below 14 K solely comes from the junction. The junction resistance further drops to zero in two stages, as illustrated in the red and blue areas in Fig. 2b, which we conjecture are contributed by the Josephson effect from bulk and surface superconductivity, respectively. In temperature regime (I) from 14 K to 12 K, the rapid resistance drop may be attributed to the Josephson effect from bulk superconductivity. This is evidenced by almost overlapping resistance curves under zero-field cooling (ZFC) and 1 T field cooling (FC). FTS is a type-II superconductor with a large upper critical field $H_{c2}$ well above 50 T[22]; hence in this bulk state-dominated regime, the resistance shows little response to 1 T magnetic field. Below 12 K (temperature regime II in Fig. 2b), the resistance further drops to zero upon zero-field cooling. In contrast, the Josephson supercurrent is suppressed under 1 T field cooling, indicating a different superconducting order with distinctive magnetic response compared to the bulk states in regime I. Hence the lower temperature transition in regime II can be tentatively attributed to the surface superconducting states, as the lower $T_c$ of surface superconductivity is directly confirmed by a smaller superconducting gap on the surface (1.8 meV) versus in the bulk (2.5 meV) through high-resolution ARPES measurements[11]. At a base temperature of 2 K, a critical current $I_c$ of 290 μA is measured. Our device is in the strongly overdamped limit since no apparent hysteresis is observable in the I–V curve.

## Concurrent ferromagnetic and superconducting orderings

Figure 3a shows the magnetic field response of a representative vJJ device, where a magnetic hysteresis behavior is observed in the junction resistance at 2 K, suggesting an unexpected ferromagnetic order arises in this vJJ structure. The junction resistance drops from a finite value to zero only when the field passes zero, and the center of the

superconducting window appears around ±30 mT. We define the center of the zero-resistance window as the coercive field of the emerging ferromagnetism. We noticed pronounced resistance jumps and noises during the magnetic field sweeping, which is most likely related to random flux dynamics during the domain switching near the coercive field. Such jumps are suppressed at higher magnetic fields where all the domains are aligned with the external magnetic field. The noises also do not exist in the time domain if we pause the magnetic field at a certain value. Such magnetic hysteresis behavior is reproducible in other additional devices we fabricated (see additional data in Supplementary Note 2). Another proof of unconventional ferromagnetism amidst superconductivity is the magnetic history dependence on the critical current. The $I_c$ in a zero-field-cooled device will be significantly reduced after applying a "magnetic pulse", i.e., experiencing a sufficiently large magnetic field bias (e.g., 100 mT) and then returning to zero field (See Supplementary Note 3). After removing the magnetic field, the remnant magnetization will suppress the superconducting order parameter $\Delta_{sc}$ and thus reduce the $I_c$. Such suppression of $\Delta_{sc}$ is reversible by elevating the temperature above the Curie temperature of this ferromagnetic ordering to demagnetize the device and then cool it back to the base temperature (see Supplementary Note 3). Here the Curie temperature is roughly estimated from the temperature above which the magnetic hysteresis loop vanishes (see Fig. S6 and Supplementary Note 4). We find the demagnetization temperature around 12 K, which coincides with the superconducting critical temperature $T_c$, implying that these two orderings may be strongly associated The strong correlation between ferromagnetism and superconductivity in our FTS vJJ substantially differs from heterogeneous vdW magnetic Josephson junctions where ferromagnetism and superconductivity are independent and repulsive[23–25].

Evidence of the spontaneous time-reversal symmetry breaking in the FTS system has been recently reported by the laser ARPES experiment[14], wide-field nitrogen vacancy imaging[15], and Sagnac magneto-optic Kerr effect (MOKE) imaging experiments[16], but has not yet been confirmed through transport measurements in single crystals. By stacking two layers of FTS flakes, the exchange energy of magnetism may be enhanced through the hybridization between the surface states of top and bottom flakes, allowing us to directly measure the magnetic behavior in such vJJ structures. This enhancement of ferromagnetism can be explained by the spin-spin interactions between the two adjacent layers as follows. For a single FTS flake, the in-plane spin polarization direction can be arbitrary due to the $U(1) \times U(1)$ symmetry. However, the non-negligible inter-layer spin-spin interactions can further lower the ground state energy by reducing this degeneracy down to a $U(1) \times Z(2)$ symmetry. Therefore, the inter-layer

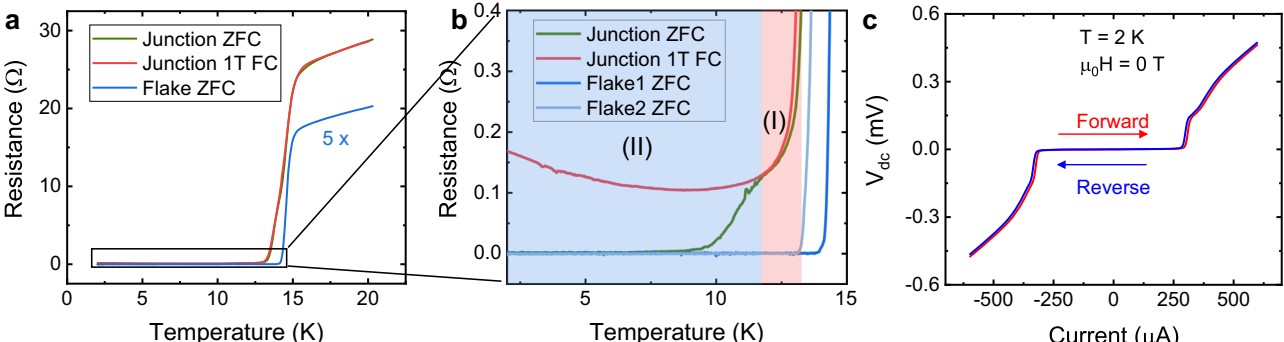

**Fig. 2 | Superconducting current in an FTS vJJ. a** Temperature versus resistance of a junction device (F8) under zero-field cooling (ZFC, gray), 1 T field cooling (1 T FC, red) and the bare FTS flake under ZFC (blue, 5 times enlarged). **b** Zoomed-in area of the superconducting states in **a**. The junction resistance drops to zero in two stages (I) and (II). **c** I–V curves of the junction measured at 2 K and 0 T. No hysteresis behavior is observed, indicating that the Josephson junction is in the overdamping regime.

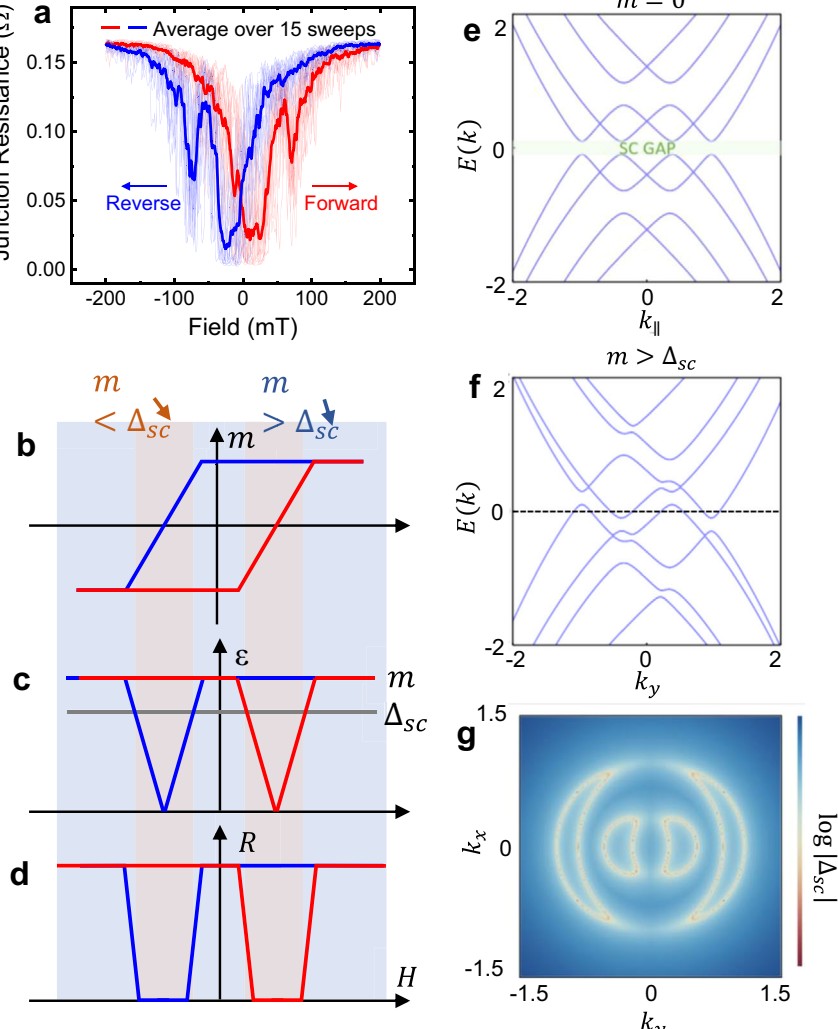

**Fig. 3 | Completing ferromagnetic (FM) and superconducting (SC) ordering in FTS vJJs. a** The magnetic hysteresis loop of the superconducting states. The dark curves show an average over 15 sweeps in the background to eliminate the contribution from flux jumps and domain dynamics. The curves were measured at 2 K with 100 μA AC current, and the magnetic field direction is parallel to the flakes, as shown in Fig. 1b. **b–d** Illustration of **b**, the magnetic moment $m$, **c** competing FM and SC order parameters: Zeeman energy $\Delta_{FM}$ and superconducting gap $\Delta_{SC}$, and (**d**) junction resistance during the magnetic field sweeping. **e, f** Superconducting band structure without and with in-plane magnetization $m_x = 0.2$. A gapless superconducting state is formed. **g** The gap function in momentum space. An arc-like energy contour from the partial Fermi surface under magnetization.

coherent spin-polarized state has much stronger magnetism. A detailed analytical understanding on how interlayer spin-spin interactions can enhance magnetism from the Ginzburg-Landau theory is discussed in Supplementary Note 5. As a result, the magnetic hysteresis behavior is discernable in such a vJJ device geometry, while it is absent in a bare FTS flake (Supplementary Note 6). We also notice that the hysteresis behavior is only observable with the presence of an in-plane magnetic field (i.e., perpendicular to the c-axis), showing a strong magnetic anisotropy with an in-plane easy axis (Supplementary Note 2). This is consistent with the reported stray field direction in the nitrogen-vacancy center magnetometry measurements[15].

Our experimental observation of magnetic hysteresis loops can be qualitatively explained by the coexistence of ferromagnetic and superconducting ordering parameters, as shown in Fig. 3b–d. A square-like hysteresis loop describes the magnetization of a ferromagnetic ordering (Fig. 3b), where the Zeeman energy of ferromagnetic ordering $\Delta_{FM}$ is minimized near the coercive field during magnetic domain switching. In this regime, the superconducting gap energy $\Delta_{SC}$ exceeds $\Delta_{FM}$ (Fig. 3c); therefore, the device reaches zero-resistance superconducting states (Fig. 3d). We also propose an

effective model Hamiltonian to describe our system. We ignore all the bulk bands that are trivially gapped out by an s-wave pairing potential below $T_c$, and study the hybridization of top and bottom surface Dirac states at the interface with the inclusion of surface magnetism $\vec{m}$. The four-by-four Hamiltonian reads:

$$H_{surf} = v_F\left(k_x\sigma_y - k_y\sigma_x\right)\tau_z + \left(t_0 + t_1 k^2\right)\sigma_0\tau_x \\ + \left(m_x\sigma_x + m_y\sigma_y + m_z\sigma_z\right)\tau_0 + \delta_\mu\sigma_0\tau_z - \mu\sigma_0\tau_0 \quad (1)$$

where $v_F$ is the Fermi velocity of surface Dirac states, $t_0$ and $t_1$ describe the hybridization strength between the top and bottom surface Dirac states, $\vec{m} = \left(m_x, m_y, m_z\right)$ is the spin magnetization, $\delta_\mu$ is the relative chemical potential shift of the top and bottom Dirac states, and $\mu$ is the overall chemical potential. Here $\sigma$ and $\tau$ are Pauli matrices for the spin and layer degrees of freedom. Without magnetization, a superconducting gap is fully open near the Fermi surface, as shown In Fig. 3e. Since the surface Hamiltonian has a continuous rotational symmetry about the z-axis, we choose $m_x = 0.2, m_y = 0$ to turn on magnetization without loss of generality. We also set $m_z = 0$ since the

experiment result suggests the magnetization mainly concentrates in the x-y plane. The competition between superconductivity and ferromagnetism produces gapless superconducting states, as shown in Fig. 3f, which give rise to an arc-like energy contour, known as partial Fermi surfaces (Fig. 3g)[26]. In this state, the superconducting nature is preserved locally in the momentum space, whereas the overall sample can exhibit finite resistance due to scattering processes (more discussion on the model Hamiltonian can be found in Supplementary Note 7). While the partial Fermi surface picture can phenomenologically interpret the observed results, we shall acknowledge that additional contributing factor such as nucleation of vortex-antivortex pairs may also potentially give rise to a finite resistance[17,27].

## Fraunhofer pattern and mixture of 0−π Josephson junctions

We next present prominent Fraunhofer oscillatory features in FTS vJJs with a multi-domain texture of mixed 0 and π phase junctions. Figure 4a shows the differential resistance mapping as a function of DC bias current and external magnetic field measured at 2 K, where "single-slit" type interference patterns are resolved, confirming the established Josephson phase-current relation in this vJJ structure. An oscillation period $\Delta B = 5.5 mT$ is extracted from the diffraction pattern, which can be translated into a junction area of 0.38 μm² under the flux quantization condition: $\Delta B \cdot S = \Phi_0 = h/2e$. Here $S$ is the effective junction cross-sectional area perpendicular to the magnetic field.

Given that the junction length (that is, the average width over the overlapping area perpendicular to the magnetic field) is ~ 7 μm, we can deduce a junction width $W = d + 2\lambda_L = 56 nm$ ($d$ and $\lambda_L$ denotes the vdW gap size and in-plane London penetration depth of FTS). With a negligible vdW gap $d < 1$ nm, we find $\lambda_L = 28 nm$ at 2 K in good agreement with the previously reported value[28,29]. The Josephson penetration depth[23,30] $\lambda_J = \sqrt{\hbar/2\mu_0 e(d + 2\lambda_L)J_c} = 9.1 \mu m(2)$ is more than one order larger than the thickness of two FTS flakes combined. ($\hbar, \mu_0, e, J_c$ are reduced Planck's constant, vacuum permeability, electron charge, and critical current density, respectively). Therefore, our device should be well within the small Josephson junction limit. Note that the Fraunhofer pattern is only observable in a small magnetic field window below the coercive field, where the long-range ferromagnetic ordering is not yet established and thus no obvious hysteresis is present (Supplementary Note 8). After experiencing large magnetic field bias, the device's critical current mapping shows hysteresis due to magnetization; meanwhile, the Fraunhofer patterns become ill-developed as they are overwhelmed by noises likely caused by flux jumps and domain motions (Supplementary Note 9).

The Fraunhofer pattern also shows the features of π Josephson junctions which infers the existence of magnetism. Near zero field, a local minimum can be observed in the Fraunhofer pattern in Fig. 4a. This is the signature of a magnetic Josephson junction with a π phase offset in the ground state. It is well known that the superconducting

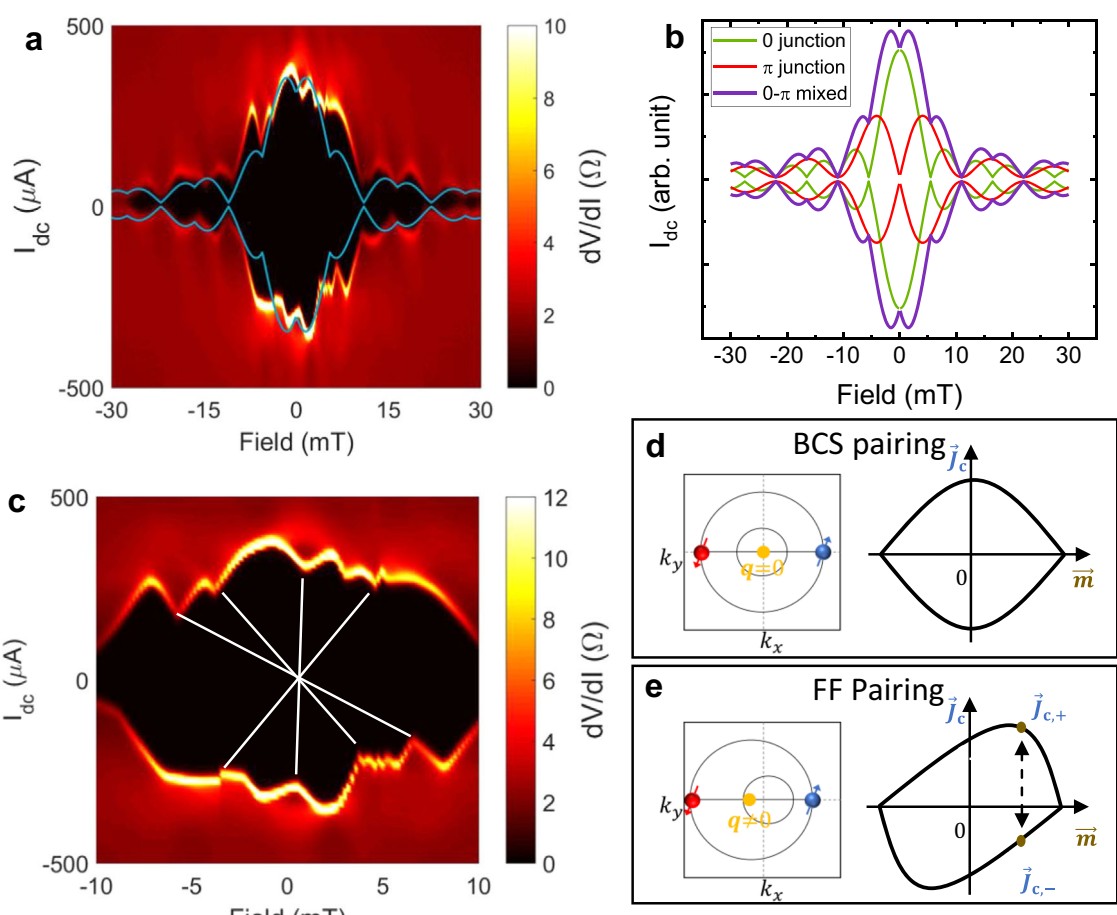

**Fig. 4 | The Fraunhofer pattern and mixture of 0−π phase in vJJs. a** Color mapping of differential resistance dV/dI as a function of magnetic field and DC current bias. Solid lines: Numerical fitting of Fraunhofer pattern with 0-π mixed junctions. All data is acquired at 2 K. **b** Oscillatory patterns of critical current with components from 0, π, and 0-π mixed junctions. **c** Zoomed-in regime at low-field mapping with skewed Fraunhofer patterns. Solid lines highlight centrosymmetric features of non-reciprocal current-field relationship described by $I_{c+}(\boldsymbol{B}) = -I_{c-}(-\boldsymbol{B})$. **d, e** Schematics of the Bardeen−Cooper−Schrieffer (BCS) pairing (**d**) and the Fulde−Ferrell (FF) pairing (**e**) mechanisms (left panels) that result in symmetric and skewed current-field relationships, respectively (right panels).

order parameter in a ferromagnetic Josephson junction will alternate between 0 and π phases by varying magnetic layer thickness[4], which can be characterized by an oscillatory sign of critical current $I_c(d_F) \sim \exp(-\frac{d_F}{\xi_{F1}})\cos(\frac{d_F - d_{dead}}{\xi_{F2}})(3)$[31]. Here $\xi_{F1,2}$ are real and imaginary parts of the complex coherence length which is very short in magnetic JJs (typically on the order of nm), and $d_F$, $d_{dead}$ are the thickness of the magnetic layer and magnetically dead layer, respectively. Due to the small superconducting coherence length in magnetic layers, the phase is extremely sensitive to the magnetic layer thickness, hence slight spatial variation (either from inhomogeneous magnetic domains or vdW gap thickness fluctuation) will create mixed domains with 0 and π phase ground states. The major features of the Fraunhofer patterns can be well reproduced by decomposing the critical current into contributions from 0 and π Josephson junctions, as indicated by the solid lines in Fig. 4a, b. We shall emphasize that the multi-domain mesoscopic structure provides a natural platform to host *1D chiral Majorana states* along domain walls with the π phase difference in superconducting order parameters[32], as predicted in the Fu-Kane model[7].

By zooming in on the low-field region in Fig. 4c, it becomes apparent that the Fraunhofer pattern is slightly skewed, and the positive ($I_{c+}$) and negative ($I_{c-}$) critical currents are bounded by a non-reciprocal field-dependent relationship $I_{c+}(\mathbf{B}) = -I_{c-}(-\mathbf{B})$. This centrosymmetric pattern is highlighted by some highly coinciding features in the opposite quadrants linked by the white solid lines (as shown in Fig. 4c). A similar skewed Fraunhofer pattern is also reported in other ferromagnetic Josephson junctions[23]. This effect is consistent with Fulde−Ferrell (FF) finite-momentum pairing states[33] arising from the ferromagnetic ordering, leading to an imbalance between opposite critical currents. Unlike traditional Bardeen−Cooper−Schrieffer (BCS) pairing (Fig. 4d), the Zeeman energy will shift the Fermi surface away from the Brillouin zone center and cause a finite center-of-mass momentum pairing (Fig. 4e). We note that the finite-momentum pairing is general in our case and can happen even without a phase transition to FF states[34]. This non-reciprocal critical current is known as

superconducting diode effect, which necessitates time-reversal symmetry breaking, and will be further discussed next.

## Stochastic field-free superconducting diode effect

The superconducting diode effect (SDE) refers to imbalanced critical current values of a superconductor in opposite DC current bias directions. SDE is premised on both lack of inversion symmetry and time-reversal symmetry[34–36]; the former is achieved by structural or crystal symmetry breaking, whereas the latter can be realized with either external magnetic field or intrinsic magnetism. So far, the majority of the reported SDEs require applying an external magnetic field. Field-free SDEs are rare to encounter (with very few recent exceptions[37–40]) due to the incompatibility of magnetism and superconductivity. In our FTS vJJ devices, the field-free SDE is observed (Fig. 5a), suggesting both inversion symmetry and time-reversal symmetry are broken in this system. The inversion symmetry can be easily broken in this stacked structure, for example, by introducing a relative chemical potential shift between top and bottom Dirac surface states (see Supplementary Note 7). The fact that SDE can be achieved without external magnetic field is a direct manifestation of spontaneous time-reversal symmetry breaking, and the diode behavior can be used to probe the interface ferromagnetism below $T_c$. More strikingly, the diode polarity can be stochastically switched by performing a thermal cycle above the superconducting $T_c$ which randomizes the spin configurations and generates random net magnetization. Two middle panels in Fig. 5a show two representative dV/dI curves measured at 2 K with zero-field before and after a thermal cycle. Here a thermal cycle refers to the procedure of heating the device to 25 K (well above $T_c$) and cooling back to base temperature of 2 K. Above $T_c$ the thermal fluctuation renders the spin configuration undetermined without established magnetism. Upon zero-field cooling, the randomized spin will freeze and trap a small but non-zero net magnetization (see Fig. 5b), the direction and amplitude of which will determine the polarity and efficiency ($\eta = \frac{I_{c+} - I_{c-}}{I_{c+} + I_{c-}}$) of the SDE. We further notice that by performing a field cooling under a moderate magnetic field of ±10 mT, the diode efficiency enhanced because of the fully aligned spin

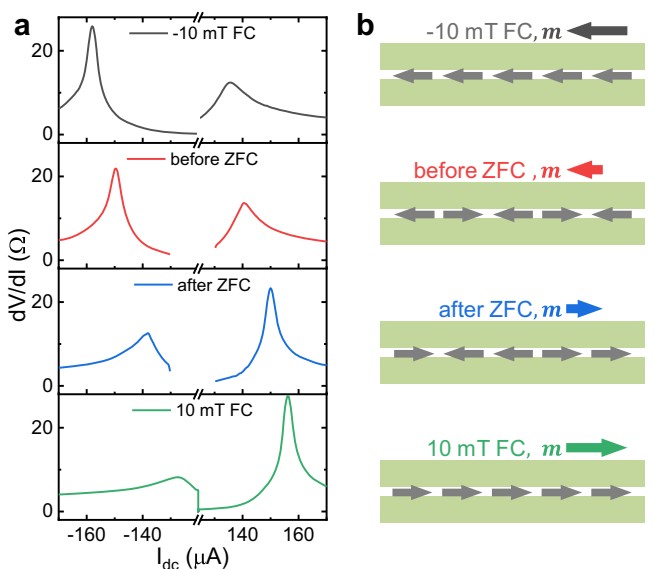

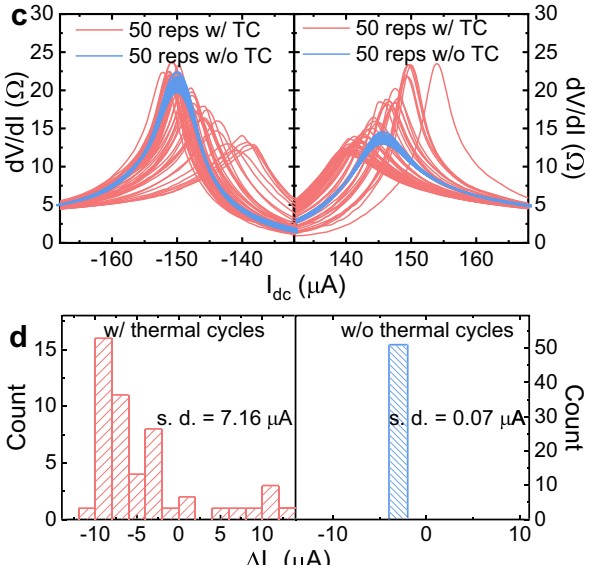

**Fig. 5 | Stochastic field-free superconducting diode effect. a** Non-reciprocal critical current measured under different magnetization conditions. All curves are measured at 2 K and zero magnetic field, but with different field cooling conditions. From top to bottom: under −10 mT field cooling (FC); before zero field cooling (ZFC); after ZFC; under +10 mT FC. **b** Schematics of spin configuration with different net magnetization describing the measurement schemes in **a. c** 50 repetitions of critical current measurements performed with (red) and without (blue) thermal cycles performed between each current scan. Here a thermal cycle (TC) refers to a procedure of heating the device to 25 K (above $T_c$) and cooling back to 2 K. **d** Histograms of stochastic critical current distribution with (left) and without (right) TCs. s.d. standard deviation.

configuration, and the polarity is deterministic upon the field direction (as shown in top and bottom panels of Fig. 5a, b). The stochastic distribution of field-free SDE, as shown by the red curves in Fig. 5c, is statistically investigated by repeating thermal cycling 50 times. In another 50 zero-field current scans without thermal cycling, the diode behavior remains unchanged (blue curves in Fig. 5c). The histograms show that the diode effect is widely distributed between thermal cycles (left panel in Fig. 5d and Supplementary Note 10). In contrast, without thermal cycling, the magnetic domains freeze and do not evolve with time (Fig. 5d, right panel; and Supplementary Note 10). The stochasticity also rules out a trivial mechanism of SDE due to the trapped flux from the superconducting coil, as the magnet was kept in the persistent mode throughout the entire measurement and thus the remnant field is fixed and should not induce such stochasticity (more discussion see Supplementary Note 10). We emphasize that if sample inhomogeneity causes magnetic regions to be completely separated from the superconducting regions across the surface, there will be no superconducting diode effect at zero magnetic field. Thus, our finding of field-free superconducting diode effect not only shows the coexistence, but also the coupling between ferromagnetism and superconductivity in FTS.

## Discussion

Here we briefly discuss the relationship between ferromagnetism and superconductivity in our FTS samples. While the previous studies suggest the coexistence of ferromagnetic and superconducting orderings[14–17], the correlation between them is yet elusive. It should be acknowledged that a small amount of excess iron atoms may exist in superconducting FTS samples and contribute to the ferromagnetism[41]. However, detailed study revealed that magnetic impurity distribution highly overlaps with non-superconducting region[21], suggesting the ferromagnetism induced by excessive iron is locally incompatible with superconductivity. In addition, such ferromagnetic ordering from iron atoms should sustain above the superconducting temperature, which is contradictory to our experimental observation and recent MOKE results[16]. This may be indicative of other more complex origins, for example, unconventional superconducting pairing mechanisms with *spontaneous* time-reversal symmetry breaking, are possible given the strongly synchronized behaviors between ferromagnetism and superconductivity. Further investigations combining high-quality transport devices and precise magnetic probes are needed to confirm such a picture.

In conclusion, we report an emergent ferromagnetism with superconductivity at the interface of Fe(Te,Se) flakes through transport measurements in the van der Waals Josephson junction device geometry. The concurrency of ferromagnetism and superconductivity suggests the close tie between these two orderings which can arise from unconventional pairing mechanisms with interface coupling. A field-free stochastic superconducting diode effect is further discovered, implying spontaneous time-reversal symmetry breaking in the superconducting states. The stochasticity in the polarity and efficiency of SDE reflects the randomness of net magnetic moment of the surface magnetism without external magnetic field. Our work provides a comprehensive understanding of the interplay between superconducting and ferromagnetic orderings and lays a foundation for understanding the TSC nature and related Majorana physics in the iron-based high $T_c$ superconductor platform.

## Methods

### Crystal growth

For the Fe(Te$_{0.58}$Se$_{0.42}$) single crystal growth, Fe (99.9%; Alfa Aesar), Te (99.5%; Alfa Aesar), and Se(99.5%; Alfa Aesar) powders with a nominal composition of 1:0.5:0.5 were thoroughly mixed, pressurized into pellets under 2000 psi, sealed into double evacuated sealed quartz ampoule, and annealed at 450 °C for 8 h and at 1050 °C for 20 h and at 750 °C for 140 h, followed by furnace cooling down to room temperature. The X-ray diffraction (XRD) experiments were carried out to confirm the crystal structure and concluded the $c$ lattice constant of 6.04 Å, which is consistent with the value reported in references [42,43]. The atomic content is determined by XRD peak position and from Energy Dispersive X-ray Spectrometry (EDS) analysis, as shown in Supplementary Note 1. The single crystal with 42% Se content falls into the topologically non-trivial phase with band inversion[21,44].

### Device fabrication

The device was fabricated using the bottom contact scheme to minimize the air exposure time during fabrication. 5/95-nm-thick Cr/Au bottom electrodes were first deposited onto SiO$_2$/Si substrate using photolithography and electron beam evaporation. Fe(Te,Se) and h-BN flakes were exfoliated from bulk crystal onto silicon substrates using scotch tapes, and then stacked together with a combination of polydimethylsiloxane (PDMS) and polypropylene carbonate (PPC) polymer films following the standard dry transfer technique[45]. The transfer/stack procedure was carried out with a home-built transfer stage inside a glove box with H$_2$O and O$_2$ levels maintained below 1 ppm.

### Material characterization

Electron transparent cross-sectional samples were prepared with an FEI Helios NanoLab 660 DualBeam (SEM/FIB). An FEI Titan 80–300 probe-corrected STEM/TEM microscope operating at 300 keV was employed to conduct atomic resolution annular dark-field scanning transmission electron microscopy (ADF-STEM) imaging analysis, with a probe convergence semi-angle of 14 mrad and a collection angle of 34–195 mrad. The bottom FTS flake was tilted to [100] zone-axis to make the interface of the two (001) oriented flakes parallel to the transmission electron beam. Atomic resolution ADF-STEM images were then taken to measure the physical vdW gap between the two flakes.

### Transport measurements

Magneto-transport measurements were performed in a Quantum Design Physical Property Measurement System (PPMS) with a superconducting coil capable of producing up to 9 Tesla magnetic field. The data was taken using the standard low-frequency (<20 Hz) lock-in technique with an AC current excitation of 10 μA at a base temperature of 2 K, unless otherwise specified. For differential resistance measurement, a DC current bias is applied via a Keithley 6221 Current source meter; the AC current/voltages are sourced/measured by SR830 lock-in amplifiers. The critical current is always read off from the outbound branch of the current scan, i.e., from 0 to a large current, to eliminate the Joule heating effect. During the dV/dI measurement, the AC excitation is kept below 0.5% of the full DC scan range. For zero-field cooling procedures, prior to cooling down, the superconducting coil was set to oscillate to 0 field to minimize the trapped flux and therefore the remnant field generated from the coil.

## Data availability

All the data that supports the plots within this paper and other findings of this study are available from the corresponding author upon request.

## Code availability

The custom codes generated during this study are available from the corresponding author on request.

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

## Acknowledgements

This work was supported by the NSF under Grants No. 1936383 and No. 2040737, the U.S. Army Research Office MURI program under Grants No. W911NF-20- 2-0166 and No. W911NF-16-1-0472. C.E. is an employee of Fibertek, Inc. and performs in support of Contract No.W15P7T19D0038, Delivery Order W911-QX- 20-F-0023. H.Z. acknowledges support from the U.S. Department of Commerce, National Institute of Standards and Technology under the financial assistance awards 70NANB22H101. A.V.D. and S.K. acknowledge support through the Material Genome Initiative funding allocated to the National Institute of Standards and Technology. The views expressed are those of the authors and do not reflect the official policy or position of the Department of Defense or the US government. The identification of any commercial product or tradename does not imply endorsement or recommendation by Fibertek Inc.

## Author contributions

G.Q., H.Y.Y. and K.L.W. conceived the project and designed the experiments. Y.L. synthesized the material. H.Y.Y. and G.Q. fabricated

the devices. G. Q. and H.Y.Y. performed the magneto-transport experiments. G.Q., H.Y.Y., C.E. and P.D. analyzed the transport data. L.H. and R.Z. provided theoretical support and discussion. H.Z., C.C., S.K. and A.V.D. carried out material characterization including XRD, TEM, EDS, and structural analysis. G.Q. and K.L.W. wrote the manuscript. All authors participated in the preparation of the manuscript and commented on the paper.

## Competing interests

The authors declare no competing interests.
