## [Peer Review File · Nature Communications]

Emergent Ferromagnetism with Superconductivity in Fe(Te,Se) van der Waals Josephson JunctionsREVIEWER COMMENTS

Reviewer #1 (Remarks to the Author):

The authors use a special material Fe (Te_{0.58}Se_{0.42}) (FTS) to construct a van der Waals (vdw) Josephson junction and successfully demonstrated Josephson coupling. Due to the ferromagnetism and superconductivity that exist on the surface of FTS at low temperatures, some exotic and interesting phenomena have been observed. It is an interesting work. Here, I have some concerns for the authors to address.

1. Line 97, "see Methods for more discussion", please present more detailed discussion about using ADF-STEM to analyze the van der Waals gap in Methods section.
2. Line 99, reference 19 is a classic work on van der Waals gap Josephson coupling, and there is no illustration on the size of gap. How do the authors consider that 2Å gap is sufficient to establish a phase separation? If the vdw gap is not sufficient to establish a phase separation, the surface of FTS flake with smaller superconducting gap acts as a different superconductor connecting two FTS flakes, which also may form a S-S'-S Josephson junction, similar to the results reported in the article of Scientific Reports, 2016, 6:35694.DOI:10.1038/srep35694.
3. Line 106, "green" should be a typo. The authors conjecture that two stages of resistance drop are contributed by the Josephson effect from bulk and surface superconductivity. The blue curve in Fig. 2b refers to the R-T behavior of one FTS flake, but what about the other FTS flake's R-T curve? Since the flakes with different thickness have different superconducting T_c, the resistance drop in regime (1) may be attributed to the combination of two processes of superconductor-normal metal transition. It's careful to say the stage of resistance drop in regime (1) may be contributed by the Josephson effect from bulk superconductivity.
4. Line 127, the R-B curve in Fig. 3a resembles a superconducting spin valve behavior. The authors should add the applied current. In addition, a platform feature can be observed in the zero-resistance window, which could be the different coercive fields of the two FTS flakes.
5. Line 143. Where does the threshold temperature for demagnetization come from? Only one description about demagnetization process is in Supplementary Note 3, and the temperature is 25 K, which is different from the superconducting temperature T_c of 12 K or 14K.
6. Line 157, "This enhancement of ferromagnetism can be explained by the spin-spin interactions between the two adjacent layers". Please cite some literatures.
7. Line 192. The authors use a model to explain the formation of the partial Fermi surface and the tiny but finite resistance. There could be other complex reasons for the tiny resistance, such as the spontaneous nucleation of vortex and anti-vortex pairs giving a non-vanishing resistance. Please clarify it.
8. Line 203. S should be the effective junction cross-sectional area.
9. Line 211. The Josephson penetration depth is much larger than the junction's size. We can get a conclusion that the device should be a "small" junction, which means the current flow across the junction should be uniform spatially. There is a short description of the Josephson penetration depth in this article (10.1038/s41467-021-26946-w).
10. Supplementary Note 9. The stochastic average values of positive and negative critical current reflected in Figure S11b make no sense in illustrating a net remnant magnetic field trapped from the superconducting coil. The plot of ΔI_c vs. Repetition can be more reasonable, illustrating the polarity of SDE simultaneously.

Reviewer #2 (Remarks to the Author):

In the manuscript titled "Emergent Ferromagnetism with Superconductivity in Fe(Te,Se) van der Waals Josephson Junctions" by Qiu et al., the authors present experimental evidence demonstrating the presence of spontaneous time-reversal symmetry-breaking superconducting states in FeTeSe van der Waals Josephson junction devices. FeTeSe has gained attention as a potential candidate for hosting topological superconductivity through a self-proximity effect. However, the evidence of unconventional superconductivity thus far has been somewhat elusive, primarily relying on localized scanning probe measurements. The key contribution of this paper lies in the discovery of coexisting ferromagnetism and superconductivity, which would typically be suppressed in a conventional s-wave scenario, thereby indicating the presence of unconventional superconductivity. The switchable field-free superconducting diode effect is fairly new. The van der Waals Josephson junction structure is also somewhat unique in probing the surface magnetism in this material. Overall I find the manuscript and its associated data to be of high quality; and the results can convincingly support the argument. While the manuscript is in general suitable for publishing on Nature Communications, a few minor points should be addressed:

(1) FeTeSe is known to be extremely air-sensitive. Surface oxidization may introduce an iron-oxide layer (magnetic). How can the author rule out this possibility?

(2) Following the previous comment, the inhomogeneity and excess iron atoms in FeTeSe may contribute to similar magnetism. The authors need to differentiate their conclusion from this trivial case.

(3) The stochastic field-free superconducting diode effect is an interesting result and supports the argument of spontaneous TRS breaking. The statistic distribution in Fig. 5c and 5d can be explained by magnetic moment reset by thermal cycles. However, there seems to be a systematic offset in the average value of ΔI_c in both panels in Fig. 5d. This cannot be explained by the current picture. Can you elaborate?

Response to Reviewers

Reviewer #1 (Remarks to the Author):

The authors use a special material Fe (Te_{0.58}Se_{0.42}) (FTS) to construct a van der Waals (vdw) Josephson junction and successfully demonstrated Josephson coupling. Due to the ferromagnetism and superconductivity that exist on the surface of FTS at low temperatures, some exotic and interesting phenomena have been observed. It is an interesting work. Here, I have some concerns for the authors to address.

1. Line 97, “see Methods for more discussion”, please present more detailed discussion about using ADF-STEM to analyze the van der Waals gap in Methods section.

Response: In the revised manuscript, we analyzed the van der Waals gap with a more quantitative method. The line intensity of the ADF-STEM image was integrated as shown in Fig. 1e and Fig. R1. An interatomic gap of 3.6 Å was determined from the two peaks across the van der Waals gap. We also included detailed information on ADF-STEM technique in the Method part.

In the revised manuscript:

Atomic resolution annular dark-field scanning transmission electron microscopy (ADF-STEM) imaging of the cross-sectional sample (Fig 1c and 1d) reveals an interatomic vdW gap of 3.6 Å between the two FTS flakes, as shown in Fig. 1e (see Methods for more details).

Methods - Material Characterization:

Electron transparent cross-sectional samples were prepared with an FEI Helios NanoLab 660 DualBeam (SEM/FIB). An FEI Titan 80-300 probe-corrected STEM/TEM microscope operating at 300 keV was employed to conduct atomic resolution annular dark-field scanning transmission electron microscopy (ADF-STEM) imaging analysis, with a probe convergence semi-angle of 14 mrad and a collection angle of 34-195 mrad. The bottom FTS flake was tilted to [100] zone-axis to make the interface of the two (001) orientated flakes parallel to the transmission electron beam. Atomic resolution ADF-STEM images were then taken to measure the physical vdW gap between the two flakes.

Figure R1| High resolution ADF-STEM image of the vdW gap between two FTS flakes and line intensity integration that determines the vdW gap to be 3.6 Å.

2. Line 99, reference 19 is a classic work on van der Waals gap Josephson coupling, and there is no illustration on the size of gap. How do the authors consider that 2Å gap is sufficient to establish a phase separation? If the vdw gap is not sufficient to establish a phase separation, the surface of FTS flake with smaller superconducting gap acts as a

different superconductor connecting two FTS flakes, which also may form a S-S'-S Josephson junction, similar to the results reported in the article of Scientific Reports, 2016, 6:35694.DOI:10.1038/srep35694.

Response: While we acknowledge the reviewer's concern that the non-superconducting gap in a van der Waals Josephson junction is much shorter compared conventional Josephson junctions, our transport results and other reports on similar device structures (ref. 19) have otherwise indicated that a van der Waals gap of a few Angstroms is sufficient to form a Josephson junction with phase separation. In our experiment, the establishment of a phase difference between two FTS superconductors is primarily evidenced by the observation of Fraunhofer patterns in Fig. 4, which reflects the current-phase relationship of the DC Josephson effect. Similarly, Fraunhofer patterns were also reported in NbSe₂/NbSe₂ van der Waals Josephson junctions with a sub-nanometer gap size (ref. 19). In ref. 20, in vJJs made with cuprates, Shapiro steps were further observed in AC measurements to confirm the AC Josephson effect. All these works suggested a van der Waals gap of a few Å between two pieces of superconductors is sufficient to form a Josephson junction, provided that the evidence of phase-current relationships such as Fraunhofer patterns or Shapiro steps are observed. We would also like to address that, in all these vJJ devices, the crystalline orientations between top and bottom flakes are misaligned, and the momentum mismatch between two layers may facilitate the phase separation by obstructing Cooper pair tunneling events as also pointed out in ref. 19.

3. Line 106, "green" should be a typo. The authors conjecture that two stages of resistance drop are contributed by the Josephson effect from bulk and surface superconductivity. The blue curve in Fig. 2b refers to the R-T behavior of one FTS flake, but what about the other FTS flake's R-T curve? Since the flakes with different thickness have different superconducting T_c, the resistance drop in regime (1) may be attributed to the combination of two processes of superconductor-normal metal transition. It's careful to say the stage of resistance drop in regime (1) may be contributed by the Josephson effect from bulk superconductivity.

Response: The authors thank the reviewer for providing an alternative explanation for the two-stage superconducting transition in Fig. 1b. To clarify the origin of the two-step resistance drop, we re-plotted Fig. 1b by including the resistance of the second FTS flake (see Fig. R2). As the reviewer suggested, two flakes have slightly different critical temperatures due to the different flake thickness and/or material homogeneity. However, the transitions of two flakes are very sharp, which are different from the junction resistance that exhibits a long tail from 12 K to 8 K. Thus, we can conclude that the two-stage transition does not originate from slightly different T_c of two superconducting banks, but rather an intrinsic behavior from the junction itself.

In the revised manuscript, the R-T curves of the second flake has been added in Fig. 1b and the typo in Line 106 has been corrected.

Figure R2| R-T curves of the vdW JJ and two FTS flakes.

4. Line 127, the R-B curve in Fig. 3a resembles a superconducting spin valve behavior. The authors should add the applied current. In addition, a platform feature can be observed in

the zero-resistance window, which could be the different coercive fields of the two FTS flakes.

Response: The R-B curve is measured with 100 μA of AC current excitation, and this information has been included in the revised manuscript. The authors agree with the reviewer that the R-B curve of the junction phenomenologically resembles a superconducting spin valve behavior reported in an F/S/F structure (for example, PRL, 110, 097001, 2013). A typical superconducting spin valve structure includes a single superconductor sandwiched by two ferromagnets with different coercive fields. However, we argue that our device structure is different from a superconducting spin valve, but rather close to an S/F/S structure. For each individual flake, no magnetic hysteresis or coercive field can be directly measured from the transport (See Fig. R3). In fact, FTS flakes exhibit robust superconductivity with large in-plane upper critical field up to 50 T (Eur. Phys. J. B, 79, 289, 2011). We therefore speculate that the observation of hysteresis behavior arises from a single enhanced ferromagnetic ordering when two interfacial states are coupled, as will be further discussed in comment #6.

Figure R3| Magnetic field vs vJJ and FTS flakes resistance. The FTS flakes that serve as superconducting reservoirs exhibit no magnetic hysteresis behavior and remain in superconducting states throughout the field scanning window.

5. Line 143. Where does the threshold temperature for demagnetization come from? Only one description about demagnetization process is in Supplementary Note 3, and the temperature is 25 K, which is different from the superconducting temperature T_c of 12 K or 14K.

Response: The demagnetization temperature is determined by examining the vanishing of magnetic hysteresis, which is around 10-12 K as shown in Supplementary Note 4. This temperature coincides with the critical temperature of the Josephson junction (Fig. 2b), and is similar to the other reported onset temperature of magnetic ordering in FTS observed by other techniques such as ARPES and Sagnac MOKE in the literatures (ref. 14-16). During thermal cycles we heat the sample to 25 K which is substantially higher than the critical temperature to ensure all vortices are suppressed in the sample and hence no magnetization will be inherited from the previous cooldown.

In the revised manuscript, we clarified how the demagnetization temperature was deduced:

Here the Curie temperature is roughly estimated from the temperature above which the magnetic hysteresis loop vanishes (see Fig. S6 and Supplementary Note 4). We find the demagnetization temperature around 12 K, which coincides with the superconducting critical temperature T_c , implying that these two orderings may be strongly associated.

6. Line 157, “This enhancement of ferromagnetism can be explained by the spin-spin interactions between the two adjacent layers”. Please cite some literatures.

Response: A simple interpretation of interlayer spin-spin interactions enhancing magnetism can be understood from symmetry point of view and has been included in the following sentence in the manuscript:

This enhancement of ferromagnetism can be explained by the spin-spin interactions between the two adjacent layers as follows. For a single FTS flake, the in-plane spin polarization direction can be arbitrary due to the $U(1) \times U(1)$ symmetry. However, the non-negligible inter-layer spin-spin interactions can further lower the ground state energy by reducing this degeneracy down to a $U(1) \times Z(2)$ symmetry. Therefore, the inter-layer coherent spin-polarized state has much stronger magnetism. A detailed analytical understanding on how interlayer spin-spin interactions can enhance magnetism from the Ginzburg-Landau theory is discussed in Supplementary Note 5.

To extend the discussion, the spin-spin interaction between magnetization of top and bottom layers can be described by an effective Ginzburg-Landau theory,

$$F = \alpha_t(T)|\vec{M}_t|^2 + \beta_t|\vec{M}_t|^4 + \alpha_b(T)|\vec{M}_b|^2 + \beta_b|\vec{M}_b|^4 + \gamma\vec{M}_t \cdot \vec{M}_b$$

where we assume the second-order terms $\alpha_t(T) = \alpha_b(T) = \alpha_0 \left(\frac{T}{T_M} - 1 \right)$ with $\alpha_0 > 0$, and the fourth-order terms coefficients $\beta_t = \beta_b = \beta_0 > 0$. It also includes a bilinear coupling term, which is anti-ferromagnetic coupling ($\gamma > 0$) or ferromagnetic type ($\gamma < 0$), depending on the material details. The transition temperature is T_M in the $\gamma = 0$ limit for each layer. Remarkably, this term could generally enhance the total magnetization (e.g, enhancing the transition temperature), like the Josephson effect. To understand this, we can take the symmetric solution $|\vec{M}_t| = |\vec{M}_b| = M_0$, and then the free energy becomes,

$$F = 2\alpha_0 \left(\frac{T}{T_M} - 1 \right) M_0^2 + 2\beta_0 M_0^4 + \gamma \cos \phi M_0^2$$

where the angle ϕ is the relative magnetization angle difference between top and bottom layers (as indicated in our experiments, magnetization is almost polarized in the $x - y$ plane). Therefore, the γ term renormalizes the second-order term that determines the transition temperature as

$$2\alpha_0 \left(\frac{T}{T_M} - 1 \right) + \gamma \cos \phi \rightarrow (2\alpha_0 - \gamma \cos \phi) \left(\frac{T}{T'_M} - 1 \right)$$

where the renormalized transition temperature $T'_M = T_M \left(1 - \frac{\gamma \cos \phi}{2\alpha_0}\right)$. Therefore, a negative $\gamma \cos \phi$ will enhance the transition temperature. In addition, the enhance magnetization at zero temperature can be obtained by

$$(2\alpha_0 - \gamma \cos \phi)M_0 + 2\beta_0 M_0^3 = 0$$

which gives rise to $M_0 = \sqrt{-\frac{2\alpha_0 - \gamma \cos \phi}{2\beta_0}}$. Notice that $\alpha_0 > 0$, $\beta_0 > 0$ and $\gamma \cos \phi < 0$, thus, this explains the enhancement of magnetization via the inter-layer spin-spin interactions, which is generally consistent with the symmetry argument mentioned in the main text. The reducing of symmetry leads to a lower ground-state free energy.

The above discussion has been included in the revised Supplementary Information as Supplementary Note 5.

7. Line 192. The authors use a model to explain the formation of the partial Fermi surface and the tiny but finite resistance. There could be other complex reasons for the tiny resistance, such as the spontaneous nucleation of vortex and anti-vortex pairs giving a non-vanishing resistance. Please clarify it.

Response: The authors appreciate the reviewer's insightful suggestions on possible complex origin for the finite resistance observed in our experiments. While the model proposed in our study offers a comprehensive framework for understanding the formation of the partial Fermi surface, we recognize that the physical system under investigation is intricate. As the reviewer pointed out, the spontaneous nucleation of vortex and anti-vortex pairs is another plausible mechanism that can lead to the observed non-zero resistance. These pairs can disrupt the flow of charge carriers and introduce scattering events, thereby affecting the overall resistance behavior. Therefore we acknowledge that additional factors may also contribute to the observed finite resistance and we clarified that in our revised manuscript:

While the partial Fermi surface picture can phenomenologically interpret the observed results, we shall acknowledge that additional contributing factor such as nucleation of vortex-antivortex pairs may also potentially give rise to a finite resistance^{17,27}.

8. Line 203. S should be the effective junction cross-sectional area.

Response: The authors thank the reviewer for the rigorous definition of the junction area. We have rephrased the sentence in the revised manuscript per suggestion:

Here S is the effective junction cross-sectional area perpendicular to the magnetic field.

9. Line 211. The Josephson penetration depth is much larger than the junction's size. We can get a conclusion that the device should be a "small" junction, which means the current flow across the junction should be uniform spatially. There is a short description of the Josephson penetration depth in this article (10.1038/s41467-021-26946-w).

Response: The authors appreciate the author's comments, and the reference has been included in the revised manuscript:

With a negligible vdW gap d less than 1 nm, we find $\lambda_L = 28$ nm at 2 K in good agreement with the previously reported value^{28,29}. The Josephson penetration depth^{23,30} $\lambda_J = \sqrt{\hbar/2\mu_0 e(d + 2\lambda_L)J_c} = 9.1$ μm is more than one order larger than the thickness of two FTS flakes combined. (\hbar , μ_0 , e , J_c are reduced Planck's constant, vacuum permeability, electron charge, and critical current density, respectively). Therefore, our device should be well within the small Josephson junction limit.

10. Supplementary Note 9. The stochastic average values of positive and negative critical current reflected in Figure S11b make no sense in illustrating a net remnant magnetic field trapped from the superconducting coil. The plot of ΔI_c vs. Repetition can be more reasonable, illustrating the polarity of SDE simultaneously.

Response: The authors agree with the reviewer's comments and have replaced Fig. S11b with a plot of ΔI_c vs. Repetition (See Fig. R4). As we can see from Fig. R4, I_c traces

predominantly (41 out of 50) exhibit a negative diode effect with $\Delta I_c < 0$, which may be due to the remnant field from the vortices trapped in the superconducting magnet coil. Another observation is that there is no diminishing of field-free SDE over time (or over repetition), suggesting this is a robust and reproducible effect.

Figure R4| The difference between positive and negative critical current ($\Delta I_c = I_{c+} - I_{c-}$) vs. repetition.

Reviewer #2 (Remarks to the Author):

In the manuscript titled "Emergent Ferromagnetism with Superconductivity in Fe(Te,Se) van der Waals Josephson Junctions" by Qiu et al., the authors present experimental evidence demonstrating the presence of spontaneous time-reversal symmetry-breaking

superconducting states in FeTeSe van der Waals Josephson junction devices. FeTeSe has gained attention as a potential candidate for hosting topological superconductivity through a self-proximity effect. However, the evidence of unconventional superconductivity thus far has been somewhat elusive, primarily relying on localized scanning probe measurements. The key contribution of this paper lies in the discovery of coexisting ferromagnetism and superconductivity, which would typically be suppressed in a conventional s-wave scenario, thereby indicating the presence of unconventional superconductivity. The switchable field-free superconducting diode effect is fairly new. The van der Waals Josephson junction structure is also somewhat unique in probing the surface magnetism in this material. Overall I find the manuscript and its associated data to be of high quality; and the results can convincingly support the argument. While the manuscript is in general suitable for publishing on Nature Communications, a few minor points should be addressed:

(1) FeTeSe is known to be extremely air-sensitive. Surface oxidization may introduce an iron-oxide layer (magnetic). How can the author rule out this possibility?

Response: The authors appreciate the reviewer's insightful comments on alternative explanations on the origin of magnetism. We acknowledge that FTS is very air-sensitive (Physical Review B 100, 064517 (2019)). However, it should be noted that our TEM image shows an atomically sharp interface at the Josephson junction, and no obvious oxide layers are observed. Besides, the two-terminal resistances in our devices are consistently below 100 ohms at low temperatures, suggesting a limited amount of oxide formation which would lead to non-ohmic contacts with much higher resistances (for example, see PRB 100, 064517, 2019). Hence, we conclude that the surface oxidation is minimized in our devices.

In our device, a transparent oxide-free interface is achieved by taking extreme caution during the device fabrication. We used bottom electrode contacts to avoid exposure to air during the traditional fabrication process, and the assembly of the device was performed inside an Argon-filled glove box with O₂ and H₂O both below 1 ppm. The junction area

was capped with h-BN and a polymer film (polypropylene carbonate), before taking out from the glove box, and the device was exposed to air for less than 30 min before the measurement.

(2) Following the previous comment, the inhomogeneity and excess iron atoms in FeTeSe may contribute to similar magnetism. The authors need to differentiate their conclusion from this trivial case.

Response: We agree with the reviewers that interstitial iron impurities may exist in superconducting FTS samples (ref. 21 & Nat. Phys. 11, 543, 2015), which will also lead to highly unconventional superconducting states. However, it should be noted that long-range ferromagnetism and hysteresis behavior were not reported in these confirmed cases with Fe impurities. Recently, it has been observed using scanning SQUID that these excess Fe atoms tend to form anti-aligned out-of-plane vortex-antivortex pairs (PRX, 9, 011033, 2019; PRX 13, 011046, 2023); these pairs are connected by field lines and can induce a net in-plane magnetic moment when an external magnetic field is applied. The anti-aligned nature and a field-tunable in-plane moment are consistent with our observation of magnetic hysteresis that happens only under an in-plane magnetic field, and a zero-field superconducting diode effect that is tunable with an in-plane magnetic field. Based on our results, we cannot draw any definitive conclusion on the origin of the observed interfacial ferromagnetism. While our combined observation of magnetic hysteresis in junction resistance, 0- π phase mixing in the Fraunhofer pattern, and a field-free superconducting diode effect unequivocally suggests that time-reversal symmetry is more likely broken intrinsically in the superconducting state of FTS, we leave the interpretation of the origin of ferromagnetism open for further investigations by combining high-quality transport devices and precise magnetic probes.

(3) The stochastic field-free superconducting diode effect is an interesting result and supports the argument of spontaneous TRS breaking. The statistic distribution in Fig. 5c and 5d can be explained by magnetic moment reset by thermal cycles. However, there

seems to be a systematic offset in the average value of ΔI_c in both panels in Fig. 5d. This cannot be explained by the current picture. Can you elaborate?

Response: The authors thank the reviewer for the positive comment. The systematic offset in ΔI_c is caused by the remnant field from the vortices trapped by the superconducting coil. This can be further clearly seen by plotting ΔI_c vs. repetition (see Fig. R4 and response to Reviewer 1's Q10)) where current traces predominantly exhibit a negative diode effect polarity. It has been recently shown that a small magnetic field of a few Oe will sufficiently induce superconducting diode effect (Phys. Rev. Lett. 131, 027001, 2023), which is comparable to the typical remnant field from a superconducting coil. While the remnant field from the superconducting magnet is a plausible explanation for the offset in the average ΔI_c , it is a fixed value throughout the measurement and should not introduce the randomness in SDE. Therefore the stochasticity should only be attributed to the intrinsic magnetic behavior of the junctions.

REVIEWERS' COMMENTS

Reviewer #1 (Remarks to the Author):

I am satisfied with the revision. Therefore, I recommend a publication.

Reviewer #2 (Remarks to the Author):

I've checked the response letter and their revised manuscript. They have addressed most of my concerns. I think that the manuscript is ready for the final publication.